# Starch/Polyaniline Biopolymer Film as Potential Intelligent Food Packaging with Colourimetric Ammonia Sensor

**DOI:** 10.3390/polym14061122

**Published:** 2022-03-11

**Authors:** Min-Rui Chia, Ishak Ahmad, Sook-Wai Phang

**Affiliations:** 1Polymer Research Centre (PORCE), Department of Chemical Sciences, Faculty of Science and Technology, Universiti Kebangsaan Malaysia, Bangi 43600, Selangor, Malaysia; minruichia98@gmail.com; 2Department of Physical Science, Faculty of Applied Sciences, Tunku Abdul Rahman University College, Setapak, Kuala Lumpur 53300, Malaysia

**Keywords:** starch, polyaniline, biopolymer film, ammonia sensor, intelligent food packaging

## Abstract

The use of petroleum-based plastics in food packaging leads to various environmental impacts, while spoilage of food and misinterpretation of food-date labelling account for food insecurity; therefore, a biopolymer capable of indicating food edibility is prepared to resolve these issues. In this research, starch/polyaniline (starch/PANI) biopolymer film was synthesised and investigated as an ammonia sensor for potential application as intelligent food packaging. FT-IR and XRD were used to confirm the composition of the biopolymer films, while UV-Vis spectrometry was applied to identify the oxidation state of PANI in emeraldine form. PANI was successfully incorporated into the starch matrix, leading to better thermal stability (TGA) but decreasing the crystallinity of the matrix (DSC). The performance of the polymer-film sensor was determined through ammonia-vapour sensitivity analysis. An obvious colour change from green to blue of starch/PANI films was observed upon exposure to the ammonia vapour. Starch/PANI 0.4% is the optimum composition, having the best sensor performance with good linearity (R^2^ = 0.9459) and precision (RSD = 8.72%), and exhibiting excellent LOD (245 ppm). Furthermore, the starch/PANI films are only selective to ammonia. Therefore, the starch/PANI films can be potentially applied as colourimetric ammonia sensors for intelligent food packaging.

## 1. Introduction

There are countless adverse environmental impacts resulting from the large-scale production of petroleum-based plastics, including the risk of global warming [1]. These highly durable polymers in the environment are regarded as hazardous waste, and there are more than 250 thousand tons of plastic pieces accumulated in the ocean, posing a detrimental effect on marine culture [2,3]. The development of biopolymers is crucial for the conservation of fossil resources, reduction of greenhouse-gas emissions, and to curb polymer pollution [4]. Starch and cellulose are biopolymers that have attracted the most attention due to their abundance, biocompatibility, nontoxicity, and simplicity of manipulation [5,6,7].

On the other hand, food-safety issues have gained much public concern and attention. The contamination of foodstuffs due to microbial or chemical reactions compromises the quality and sensorial properties of the food, which may be harmful to human health [8]. Moreover, consumers may be confused on how long the food can be stored or the exact time for uneaten food items to be disposed of due to confusing food-date labelling, leading to unnecessary food wastage [9,10]. According to the Food and Agriculture Organization (FAO), approximately one-third (1.3 billion tons) of the world’s edible food is wasted annually [11]. On the other hand, the expiry date is just a vague indication of the food condition, since the condition of the food may vary with different storage environments [12]. There is a thriving concern in green and intelligent food packaging as a food-quality indicator [13].

Intelligent food packaging is defined as materials and articles that monitor the condition of packaged food or the environment surrounding the food [14,15]. Various colourimetric sensors have been developed for intelligent food-packaging purposes, mostly using natural dye such as anthocyanin, curcumin, and alizarin as their sensor components [16,17,18]. However, natural dyes usually have limited sensitivity and their effectiveness fluctuates with their source of extraction. Additionally, the natural colourants tend to degrade when exposed to the environment and pose a risk of migration of dye into food products, compromising food safety [19,20]. Therefore, a more stable, safe, and sensitive sensor must be developed for intelligent food packaging.

Polyaniline (PANI) is a good candidate as it is simple, reliable, rapid, ecofriendly, and safe from leaching [21,22,23]. The acute toxicity towards *Rhinella arenarum* of HCl-doped PANI nanofibre has been studied through the amphibian embryo toxicity (AMPHITOX) test, where it shows no toxicological effects even at a high concentration of 400 mgL^−1^ [24]. Furthermore, PANI-H has also been tested with mouse embryonic fibroblast cells via MTT assays, showing mild cytotoxicity. In fact, the cytotoxicity of PANI is due to the residual aniline trimer, ammonium persulfate, and polar substances with low molecular weight [25]. As long as the sample is well-rinsed, removing unreacted aniline monomer, excess APS, and HCl, PANI will not impose significant toxicity and is suitable as an additive in food packaging. PANI in its emeraldine-salt form has been studied for its potential application in intelligent food packaging, acting as an electroactive pH sensor in hydroxyethyl cellulose-based film [26]. In addition, PANI’s application as a fish-freshness indicator via colourimetric method was also studied [21,27,28].

PANI has chains that are arranged in an orderly way, with phenyl rings and nitrogen-containing groups alternating with each other, as shown in Figure 1 [29]. It exists in one of the three stable oxidising states of salts and bases, which are the pernigraniline, emeraldine, and leucoemeraldine. Emeraldine in the form of salt appears green while the base form appears blue [30]. The interconversion of emeraldine salt and base oxidation states of PANI is achieved through protonation and deprotonation. Thus, the pH (presence of acid or base) will affect the oxidation state of PANI, resulting in a change of colours, rendering PANI a colourimetric sensor. Food spoilage by bacteria causes the formation of gas molecules such as trimethylamine (TMA) and dimethylacetamide (DMA), as well as decomposing urea and amino acids into ammonia [31]. Therefore, the ammonia released during microbial spoilage of food may act as a marker gas for the evaluation of food quality [31,32].

In a nutshell, starch-polymer matrix, which is biodegradable, is expected to resolve polymer pollution and to conserve fossil-fuel resources. However, their high electrical resistivity has become a stumbling block for the bio-based polymers to be adopted as a commercial food-freshness indicator. Thus, the noninvasive and leach-free PANI will be incorporated into the starch matrix for the ammonia-sensing property to indicate food freshness.

Although PANI has been widely studied for its sensor properties, based on our knowledge, there is no published research regarding the sensor performance of starch/PANI biopolymer film as a colourimetric ammonia sensor. In this study, starch/PANI polymer films were synthesised to study their chemical and physical properties as an ammonia sensor for its potential application in the intelligent food-packaging industry.

## 2. Materials and Methods

### 2.1. Materials

Ammonium persulfate (APS), aniline hydrochloride (ANI-HCl), acetone, hydrochloric acid, glycerol, and 30% ammonium hydroxide were purchased from Sigma Aldrich (St. Louis, MO, USA), while ethanol, methanol, toluene, and cassava starch were purchased from R&M Chemicals (Petaling Jaya, Malaysia), Systerm (Shah Alam, Malaysia), ChemSoln (Shah Alam, Malaysia), and Cap Kapal ABC (Georgetown, Malaysia), respectively.

### 2.2. Synthesis of Polyaniline (PANI)

#### 2.2.1. Synthesis

Firstly, 100 mL of 0.25 M ammonium persulfate (APS) was added dropwise into 100 mL of 0.2 M of aniline chloride (ANI-HCl) under 0–5 °C. Both solutions were chilled in a refrigerator for 1 h prior to the dripping. The mixture of the solutions was left to be stirred overnight. After that, the PANI precipitate was filtered and washed thrice with 0.2 M HCl to remove unreacted monomer and APS, then washed thrice with acetone to remove excess HCl dopant, oligomers, and low-molecular-weight organic intermediate. The collected PANI was dried and ground into powder.

#### 2.2.2. Characterisation

The synthesised PANI was characterised using Fourier transform infrared spectroscopy (FT-IR) (Spectrum 400, Perkin Elmer, Waltham, MA, USA) in a range of wavenumbers from 650 cm^−1^ to 4000 cm^−1^ and ultraviolet-visible (UV-Vis) spectroscopy (UV-1800, Shimadzu, Kyoto, Japan) from wavelengths 250 nm to 900 nm.

### 2.3. Fabrication of Starch/PANI Films

#### 2.3.1. Synthesis

Initially, 10 g of cassava starch, 5 g of glycerol, and the PANI (0.1, 0.2, 0.3, 0.4, 0.5 wt.%) was added into 85 mL of distilled water. The suspension was stirred and sonicated. After sonication, the suspension was heated at 70 °C with constant stirring until it was gelatinised. The coagulation was poured into a mould and dried at 70 °C for 24 h. The thickness of the starch and starch/PANI films were in the range of 0.529 mm to 0.699 mm.

#### 2.3.2. Characterisation

The films were characterised through FT-IR spectroscopy. Deconvolution was conducted to identify overlapping peaks based on the Gaussian–Lorentzian model using the OMNIC 8 software provided by Thermo Fisher Scientific Inc. (Waltham, MA, USA). Additionally, the crystallinity of the films was studied through X-ray diffraction (Bruker/D8 Advance) under the conditions of voltage 40 kV, X-ray wavelength (λ) 1.5406 nm, and current 30 mA. The diffractograms were smoothed based on a Savitzky–Golay filter using the Origin Pro 8.5 software by OriginLab Corporation (Northampton, MA, USA). Thermogravimetric analysis (TGA) and differential scanning calorimetry (DSC) were carried out to study the thermal properties of the films STA 449 F3 Jupiter, NETZSCH, Selb, Germany).

### 2.4. Sensor Performance of Starch/PANI Films

For sensor application, the starch/PANI films were exposed to the ammonia solutions (200, 400, 600, 800, 1000 ppm) in a closed system for 30 min to study the films’ sensitivity towards ammonia at different concentrations. The optimum composition of starch/PANI film was then exposed to 1000 ppm ammonia solution at 30 °C, 4 °C, and −18 °C to accurately understand the applicability of the sensor at different storage temperatures. Sensor selectivity was studied by exposing starch/PANI films to different types of volatile solvents including ethanol, methanol, toluene, and acetone.

The changes of green and blue appearances of the starch/PANI films were recorded using the ImageJ software developed by the National Institutes of Health USA. The change of colour signal of the films was calculated using the following equation:(1)$$\mathrm{Change}\;\mathrm{in}\;\mathrm{colour}\;\mathrm{signal}=\frac{\left(B/G\right)}{\left(B_{0}/G_{0}\right)}$$
where B_0_ = initial blue signal before exposure to ammonia vapour; G_0_ = initial green signal before exposure to ammonia vapour; B = blue signal after exposure to ammonia vapour; and G = green signal after exposure to ammonia vapour.

The sensitivity of the sensor was monitored based on the changes in colour signals mentioned in Equation (1) where the normalised change in colour signal was obtained by dividing the change in colour signal of the sample by the change in colour signal of control sample.

## 3. Results and Discussion

### 3.1. Characterisation of PANI

The PANI synthesised through oxidative polymerisation was characterised by identifying the functional groups present in the polymer through FT-IR analysis (Figure 2). The intensity bands at 3400 cm^−1^ and 3206 cm^−1^ correspond to the N–H groups while the bands at 2902 cm^−1^ and 2899 cm^−1^ suggest the presence of aromatic C–H groups, including CH_3_ and CH_2_ [33]. Moreover, the C=C stretching deformation in the quinoid ring and benzenoid ring are responsible for the PANI’s intensity bands at 1553 cm^−1^ and 1470 cm^−1^, respectively [34]. These bands are the characteristic peaks for protonated PANI, as is the presence of the benzenoid and quinoid rings, thus indicating that the PANI is in emeraldine-salt form [35,36]. Moreover, the intensity band at 1291 cm^−1^ and 1239 cm^−1^ arises due to the aromatic conjugated C–N stretching in the quinoid ring, whereas 1109 cm^−1^ is due to the electron-like band in the quinoid unit of doped PANI [34,37]. The C–H out-of-plane bending vibration in the polymer chain is assigned to the intensity band at 991 cm^−1^. Additionally, the C–H out-of-plane stretching deformation of the meta- and parasubstituted ring of the PANI leads to the bands at 875 cm^−1^ and 788 cm^−1^, respectively [38].

The UV-Vis spectrum of PANI is shown in Figure 3. The peak at 376 nm suggests the π–π* transition in the benzenoid ring while the shoulder peak at 457 nm indicates the polaron–π* transition in the quinoid ring [39,40]. The latter shows the bipolaron behaviour of the synthesised PANI [34]. Furthermore, the hump which was seen at up to 800 nm indicates the polaron and bipolaron transition, where it shows that there is a free charge transfer in the benzenoid ring [37]. Thus, this further proves that the PANI synthesised is in the form of emeraldine salt [39].

### 3.2. Characterisation of Starch and Starch/PANI Films

The starch/PANI films were fabricated through ex-situ polymerisation. Figure 4 shows the FT-IR spectra of the starch and starch/PANI polymer film. For the FT-IR spectrum of starch film, the broad band at 3286 cm^−1^ ranging from 3600 cm^−1^ to 3000 cm^−1^ is assigned to the stretching of hydroxyl groups in the starch, glycerol, and adsorbed water [41,42]. The intensity band at 2933 cm^−1^ and its shoulder peak at 2883 cm^−1^ are attributed to the C–H bending of the anhydrous glucose unit (AGU) and CH_2_ deformation, respectively [43]. On the other hand, the single peak at 1650 cm^−1^ is due to the H–O–H stretching vibration of water molecules that were tightly bound towards the starch granule through intermolecular hydrogen bonds [43,44]. This indicates that starch is hygroscopic in nature. Moreover, the intensity band at 1205 cm^−1^ is associated with the C–O stretching of glycerol [45]. Therefore, this indicates that the glycerol is present in the starch film as a plasticiser. The weak intensity bands at 923 cm^−1^ and 761 cm^−1^ show the presence of α-1,6-glycosidic linkage and α-1,4-glycosidic linkage, respectively [41,46].

The starch/PANI polymer films show similar FTIR spectra towards the starch film. This is in agreement with the previous studies by Azmi et al. [47] and Nand et al. [48]. The main factors for the lack of PANI’s peak and wavenumber shift in the FT-IR spectra are the very small amount of PANI in the composite-polymer film and the weak FT-IR peaks of pure PANI. Besides, the solid–liquid mixture of PANI powder and the starch suspension may not be effective [47]. There is only one peak (2847–2855 cm^−1^) in the FT-IR spectra of starch/PANI (0.2–0.5%) which may arise due to the C–H vibration in PANI [35].

The XRD diffractogram of starch and starch/PANI 0.4% polymer films is shown in Figure 5 The starch film exhibits diffraction peaks of 2θ = 12.7°, 14.1°, 16.7°, 18.3°, 19.5°, and 22.1° [43,49,50]. The peak signals of starch film are very broad, indicating that it is semicrystalline with very high amorphous content. It is speculated that during the gelatinisation of the starch, heating and mechanical stirring destroy the crystalline regions of the starch granules [51].

In addition to the characteristic peaks of starch, the XRD spectrum of starch/PANI film also shows diffraction peaks at 2θ = 20.9°, 25.7°, and 27.2°. The periodicity parallel and perpendicular to the PANI chain direction was assigned to 2θ = 20.9°, and 25.7°, respectively [39,52]. The Van der Waals attraction between phenylene-ring stakes and aliphatic chains was shown by 2θ = 25.7° [53]. The slight shift of the starch’s characteristic peaks might be due to the addition of PANI, which affects the matrix’s crystallinity. Furthermore, the addition of PANI’s peak will also interfere with the original peaks.

The thermal stability of the synthesised polymer films was evaluated through TGA analysis. The thermograms of TGA and differential thermogravimetry (DTG) are shown in Figure 6. The stages and temperatures of decompositions are compiled in Table 1. Deconvolution was also carried out based on the Lorentzian model to identify weak DTG peaks [54].

The starch-polymer film exhibits three decomposition stages. The first decomposition stage at 120 °C is due to the loss of moisture on the surface of the polymer film and the water absorbed in the polymer film [55]. The main factor for the high moisture content in the starch film is the hygroscopic property of starch granules because the hydroxyl groups of amylose and amylopectin interact strongly with the water molecules through hydrogen bonding [56]. The second stage of the decomposition at 274 °C is associated with the evaporation of glycerol [42,57]. Finally, the decomposition of starch-polymer chains occurs in the third stage at 316 °C [42,58].

For the starch/PANI (0.1, 0.2, 0.3, 0.4, 0.5%) films, there is a similar degradation pattern with four decomposition stages. Similar to the starch film, the first stage of decomposition (116–140 °C) is due to the evaporation of moisture. There is a new decomposition stage at (189–236 °C) which occurs during the decomposition of HCl dopant [59]. However, the TGA transition step and DTG peak for the decomposition is very weak, as the PANI content in the polymer film is extremely low. Thus, this decomposition stage is too weak to be observed in starch/PANI 0.1%. The third (277–292 °C) and fourth decomposition (317–319 °C) stages are related to the evaporation of glycerol and decomposition of starch, respectively. Although the thermo-oxidative decomposition of the PANI backbone should occur around 300 °C, it is nonetheless not observed in the thermograms [59,60]. The absence of the degradation step might be due to the overlapping degradation temperature with the starch chain and the very small amount of PANI in the polymer film.

Despite the decomposition of PANI not being detected in the thermograms, the effect of PANI on the thermal stability of the polymer films was observed. The degradation temperature of the starch/PANI-polymer matrix (317–319 °C) is slightly higher compared to starch-polymer film (316 °C), where starch/PANI 0.4% exhibits the best thermal stability (319.8 °C).

The glass-transition temperatures (T_g_) and melting temperatures (T_m_) of the starch and starch/PANI films were identified through differential scanning calorimetry (DSC) (Figure 7, Table 2). The T_g_ steps for the polymer films were in the range of 68 °C to 77 °C. It was observed that the glass-transition step became clearer as more PANI was added into the polymer matrix. Moreover, the T_g_ decreased along with the increase in PANI mass. The observed trend was due to the disruption of the starch matrix by PANI, resulting in the increased amorphous characteristics. Thus, it shows that PANI has been well-blended into the starch matrix, causing a variation in the T_g_ pattern [61].

In addition to T_g_ steps, endothermic peaks were found in the DSC thermograms of the polymer films. The endothermic peak (277–292 °C) is attributed to the T_m_ of crystalline amylopectin and co-crystallised amylose [62]. T_m_ is the temperature at which the crystalline region of the polymer turns amorphous; it is the characteristic peak for the crystalline materials [63]. The T_m_ peaks become smaller as the composition of PANI increases in the matrix. The reduction in the enthalpy of fusion (∆H_f_) in the polymer films with higher composition of PANI shows that the crystalline characteristic was suppressed due to the disruption of the interaction between the starch-polymeric chains [49,64]. Thus, the reduction in ∆H_f_ further proves that PANI was well-incorporated into the starch matrix.

### 3.3. Sensor Performance of Starch/PANI Films

#### 3.3.1. Response Time of Starch/PANI Films towards Volatile Ammonia

The sensor response against volatile ammonia in a noninvasive manner was monitored. Figure 8 shows the response of starch/PANI 0.4% film towards ammonia vapour (600 ppm), where the intensity ratio of blue and green signal (B/G) was recorded every 5 min. The two main factors that may affect the response time of the sensor are the relative humidity of the environment and the steady-state requirements for the sample bottle [21]. Nonetheless, these factors posed minimal effect towards the sensor, as the analysis was carried out in a closed and sealed sample bottle.

The B/G signal increased from 0.84 to 1.00 in the first 30 min and achieved constant B/G signal after that. The linear correlation for the concentration ranging from 0 to 30 min shows an R^2^ value of 0.9725. However, the concentration range from 0 to 25 min shows better correlation (R^2^ = 0.9939), indicating that at the 30th minute, the sensor is saturated and is reaching the upper limit of ammonia-vapour detection [65]. Therefore, the response time of 30 min was used in the following sensor-response analyses.

#### 3.3.2. Sensitivity of Starch/PANI Films as an Ammonia-Vapour Sensor with Varying PANI wt.%

The sensitivity of starch/PANI films was studied within the ammonia-concentration range of 200–1000 ppm. After the exposure of starch/PANI films towards the ammonia solutions, there is a change of colour from green to blue which can be easily detected by the naked eye, as shown in Table 3. As the composition of PANI increases from 0.1% to 0.4%, the change of the films’ colour becomes more obvious. Starch/PANI 0.1% shows lighter shades of green and blue, while starch/PANI 0.4% exhibits much more brilliant shades of green and blue, both before and after exposure to ammonia vapour. However, starch/PANI 0.5% has a dark green colour, which is tough to differentiate from the dark blue colour after exposure to the ammonia solution due to the higher concentration of PANI.

The change of the colour of the polymer films can be attributed to the mechanism of the interaction between PANI in the form of emeraldine salt and the ammonia molecules (Figure 9). The ammonia molecules tend to interact with the nitrogen atom in the emeraldine salt due to its polar and electronegative nature [66]. Upon this interaction, the ammonia molecule withdraws a proton from the –N^+^–H site, rendering it a –NH_4_^+^ group [67]. At the same time, a partial polar bond was linked between the PANI’s nitrogen atom with the hydrogen atom, converting the emeraldine salt to emeraldine base. This leads to the change of the colour of starch/PANI film from green to blue.

The colourimetric response of the starch/PANI films as an ammonia-vapour sensor was further quantitated using the RGB (Red Green Blue) model [68]. As shown in Figure 10, starch/PANI 0.4% film shows the highest sensitivity (0.004/ppm) towards exposure to all concentrations of ammonia solutions (200–1000 ppm) among all the samples. The sensor response increases from starch/PANI 0.1% (1.12–1.14) to 0.4% (1.34–1.67) and drops when the PANI content is raised to 5% (1.26–1.59). As the composition of PANI increases in the starch matrix, there are more nitrogen sites for the ammonia to adsorb onto PANI. Since there are a higher number of interactions between the ammonia and PANI, more emeraldine salts were converted to emeraldine bases, thus making the colour changes extra-vigorous and giving the sensor higher sensitivity. Nonetheless, for starch/PANI 0.5%, the composition of PANI was too high as some of the sites were not adsorbed with ammonia. In other words, while some of the emeraldine salts in starch/PANI 0.5% were converted to emeraldine bases, the excess emeraldine salts remained. After the exposure towards ammonia solution, the starch/PANI 0.5% film retains some green emeraldine salts, making the change of colour less obvious, thus reducing the sensitivity of the sensor.

On the other hand, for the starch/PANI 0.1% to 0.4%, it can be seen that the sensor response improves more significantly when the concentration increases from 200 ppm (1.12–1.34) to 1000 ppm (1.24–1.67). At lower concentrations, the sensor responses of starch/PANI 0.1% to 0.4% are similar due to the limiting factor of ammonia concentration. Therefore, the increase in PANI concentration will not largely affect the sensor response, leading to a mild increase in sensor response. At higher concentrations, there are more ammonia molecules to interact with PANI and it is no longer a limiting factor, thus resulting in a more obvious increase in sensor response.

Furthermore, it was also observed that the response of the starch/PANI films as ammonia sensors increases along with the concentration of the ammonia solution regardless of the polymer film’s composition. Referring to the mechanism in Figure 9, the higher the concentration of ammonia solution, the more ammonia molecules will be adsorbed onto the PANI, therefore converting more emeraldine salts into emeraldine base, leading to a higher sensor response. Thus, the change of the intensity of the green and blue colour signal increases along with the concentration of the ammonia solutions.

The parameters obtained from the analysis of the linear regressions (Figure 11) are tabulated in Table 4. The limit of detection (LOD) and limit of quantification (LOQ) were calculated using the equations below [69]:(2)$$\mathrm{LOD}=3.3\times \frac{\sigma_{y,b}}{m_{c}}$$
(3)$$\mathrm{LOQ}=10\times \frac{\sigma_{y,b}}{m_{c}}$$
where: *σ_y,b_* = standard deviation of pseudo-blank; *m_c_* = slope of the curve.

An F-test was carried out to indicate the significance of the regression, where a small *p*-value (<0.05) would determine that the regression does not happen by coincidence [70]. The regressions at different PANI concentrations show *p*-values which are smaller than 0.01; thus, these regressions are significant. Besides, all the starch/PANI films show good linearity in ammonia sensing with coefficient correlation (R^2^) > 0.90 [71]. The starch/PANI 0.1% to 0.4% films also show precise sensing performance with acceptable reproducibility as they have a relative standard deviation (RSD) which is lower than 10% (n = 3) [72,73,74]. Nonetheless, the performance starch/PANI 0.5% was not sufficiently precise (RSD = 12.41%) due to the high composition of PANI which affects its sensing performance. Moreover, it was found that the LOD and LOQ ranged from 145–812 ppm and 440–2459 ppm, respectively. The starch/PANI 0.4% film is the optimum composition as it shows the best sensitivity with LOD and LOQ as low as 145 ppm and 440 ppm, respectively. According to the previous study on the colourimetric sensing of ammonia vapour by the PANI film and PANI-deposited filter paper, the LOD recorded was 170 ppm and 10 ppm, respectively [21,75]. This shows that the incorporation of starch with PANI exhibits synergistic effects, which improves the ammonia-sensing capability of PANI. We speculate that the hydrophilic property of starch enhances the adsorption of ammonia onto the film, thus improving the sensing performance. Nonetheless, the sensor performance of starch/PANI is not as sensitive as PANI-deposited filter paper, which may be due to the stronger water-absorption ability of filter paper.

In general, the standard indicates that the total volatile basic nitrogen (TVB-N) should not be higher than 15,000 ppm for all types of meat, where the thresholds are higher for certain food products such as beef, fish, and crustaceans [76]. Therefore, the ability for the starch/PANI 0.4% film to detect ammonia at a concentration as low as 145 ppm is rather impressive.

#### 3.3.3. Sensitivity of Starch/PANI Film as an Ammonia-Vapour Sensor at Various Temperatures

To investigate the performance of the sensors more accurately, the starch/PANI films were exposed to 1000 ppm ammonia solution at different temperatures including 30 °C, 4 °C, and −18 °C, imitating the storage conditions under room temperature, refrigerator temperature, and freezer temperature, respectively. Since the fluctuation of the weather might affect the room temperature, the analysis test for the sensor performance was set at 30 °C in a water bath to give a constant temperature regardless of the weather. Furthermore, it was set based on the average indoor air temperature in Malaysia, at around 27 °C to 33 °C [77].

As shown in Figure 12, the starch/PANI 0.4% film shows the best sensor response of 1.67 towards the ammonia solution at a temperature of 30 °C, where the colour change of the polymer film can be easily detected with the naked eye. However, at 4 °C and −18 °C, there are no obvious colour changes of the polymer films. By using the RGB model, a slight change of colour can be detected on both polymer films by giving responses as low as 1.17 and 1.15, respectively, but the change is too negligible for them to be applied as an ammonia sensor at 4 °C and −18 °C.

At a higher temperature, the air is capable of holding more water vapour. Specifically, at 30 °C, the water-vapour-holding capacity of the air is 30 gcm^−3^, which supports better vaporisation of ammonia solution. When the temperature of the air is cooled below 20 °C, where it exceeds its dew point temperature, the water vapour begins to condense and the vaporisation of ammonia is retarded [68]. At even lower temperatures, specifically sub-zero temperatures, the water will be frozen into ice, halting the vaporisation of ammonia [78]. This is because the ammonia molecules have high affinity towards the water [79]. Thus, at 30 °C, the air-moisture content is higher and this moisture promotes the adsorption of ammonia onto the hydrophilic starch-based polymer film, resulting in a higher sensor response (1.67) [68]. At lower temperatures (4 °C and −18 °C), there is less moisture in the air to facilitate the transfer of ammonia molecules; thus, less adsorption of ammonia on the polymer films leads to a weak sensor response (1.17 and 1.15, respectively).

#### 3.3.4. Selectivity of Starch/PANI Films as Ammonia-Vapour Sensor

To investigate the selectivity of starch/PANI films towards ammonia vapour, the polymer films were exposed towards various volatile solvents including ethanol, methanol, toluene, and acetone (Figure 13). Starch/PANI film exhibited outstanding sensitivity to ammonia only, and no response towards the remaining solvents.

In addition to the reaction between the ammonia and the nitrogen of PANI, solvent solubility also plays a role in the selectivity of starch/PANI as an ammonia sensor. The solubility of the polymer matrix and solvents are estimated using the Hansen solubility parameters (HSP) (δ) which includes the solvent’s atomic dispersion forces (δ_D_), the molecular polar dipole–dipole forces (δ_P_), and molecular hydrogen bonding (δ_H_) through electron exchange [80]. The individual components will have high affinity towards each other when their HSP values are similar due to the “like dissolves like” concept [80,81]. The HSP values of the polymer matrix and volatile solvents are tabulated in Table 5. Since there are no exact published values on the HSP of the starch matrix, the HSP of dextran was used because it was presumed to be very similar to the amorphous cellulose [80]. The difference in the HSP values (∆δ) between the starch matrix (δ = 38.6 MPa^1/2^) and ammonia (δ = 33.3 MPa^1/2^) is smaller compared to the other volatile solvents (δ = 18.2–29.6 MPa^1/2^). This shows that ammonia is more miscible in the starch-polymer matrix. Thus, the starch/PANI films show the best sensitivity towards ammonia and do not show colourimetric responses to the other volatile solvents.

## 4. Conclusions

A green and biodegradable ammonia-sensing film was developed for the potential indication of food spoilage in intelligent food packaging. The starch/PANI films were synthesised through ex-situ polymerisation and the PANI was successfully incorporated into the starch matrix. The PANI increases the thermal stability of the matrix and increases the amorphous characteristic of the film. Upon the exposure towards ammonia vapour, a significant colour change of the film from green to blue was observed with good linearity and precision. Furthermore, it shows sensitivity and has an LOD at even very low concentrations of ammonia. The starch/PANI 0.4% was found to be the optimum composition of the film, showing selectivity towards ammonia vapour. Thus, the environmentally friendly and user-friendly starch/PANI film has outstanding potential to be applied in intelligent food packaging.

## Figures and Tables

**Figure 1 polymers-14-01122-f001:**
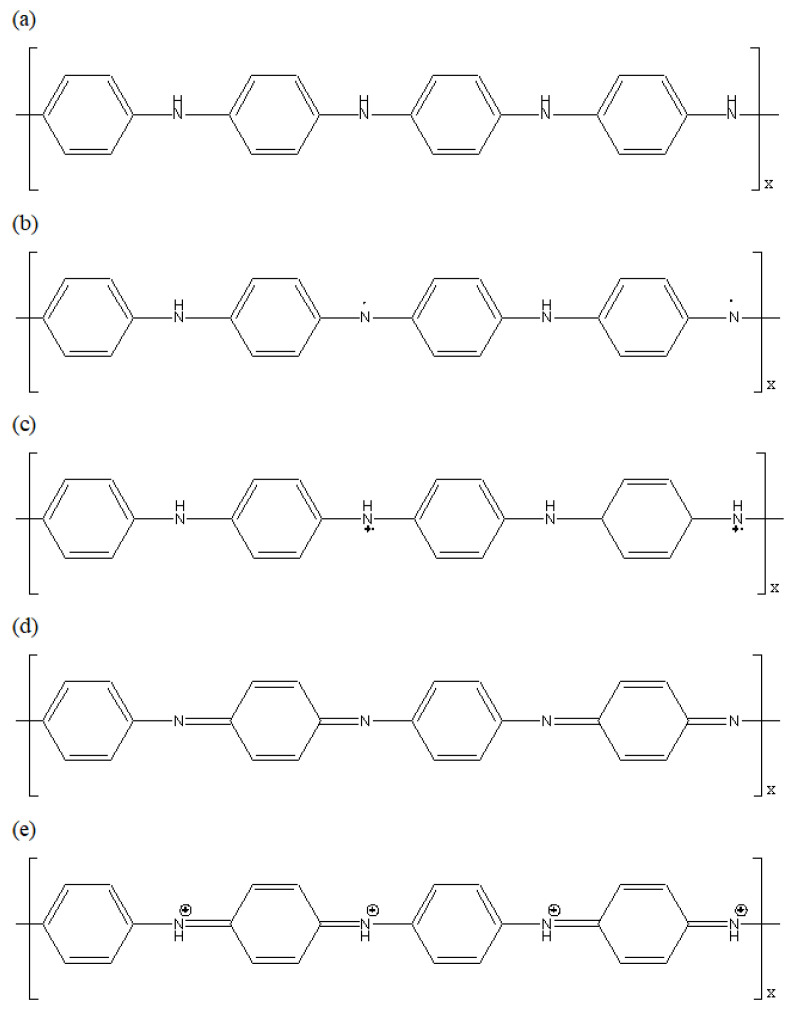
Different oxidation states of PANI. (**a**) leucoemeraldine (white/colourless); (**b**) emeraldine base (blue); (**c**) emeraldine salt (green); (**d**) pernigraniline base (violet); (**e**) pernigraniline salt (blue).

**Figure 2 polymers-14-01122-f002:**
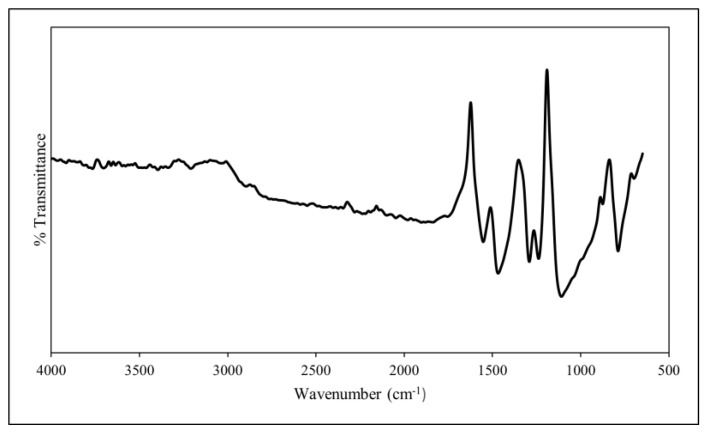
FT-IR spectrum of PANI.

**Figure 3 polymers-14-01122-f003:**
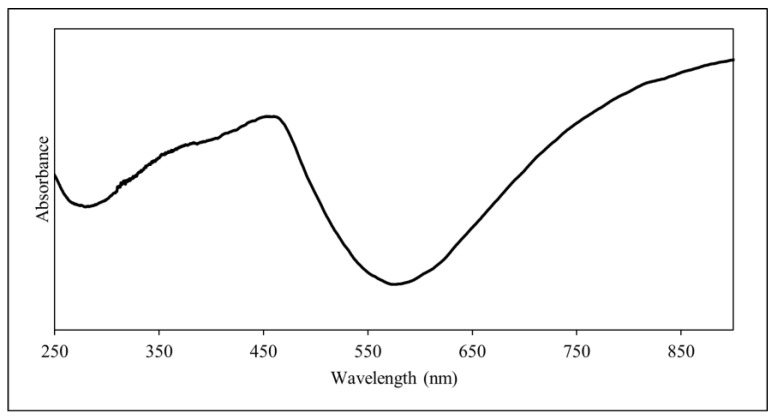
UV-Vis spectrum of PANI.

**Figure 4 polymers-14-01122-f004:**
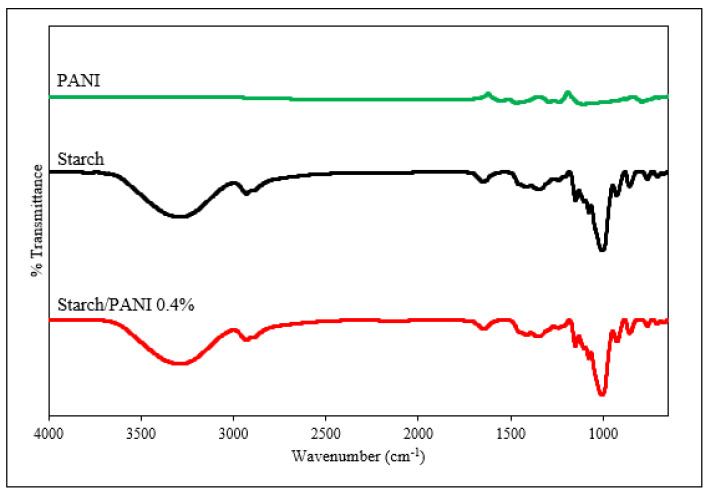
FT-IR spectra of pristine PANI, starch, and starch/PANI 0.4% polymer films.

**Figure 5 polymers-14-01122-f005:**
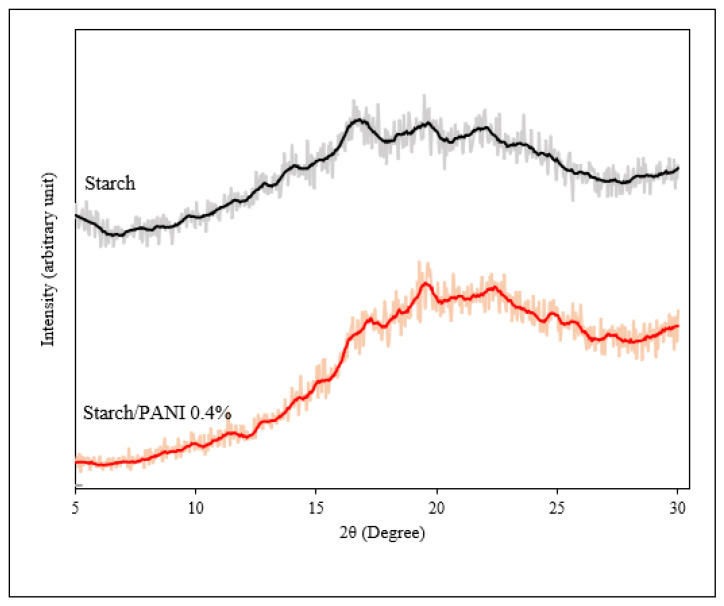
XRD diffractograms of starch and starch/PANI 0.4% polymer films (The actual experimental data and smoothed data were represented by the shadow and solid lines, respectively).

**Figure 6 polymers-14-01122-f006:**
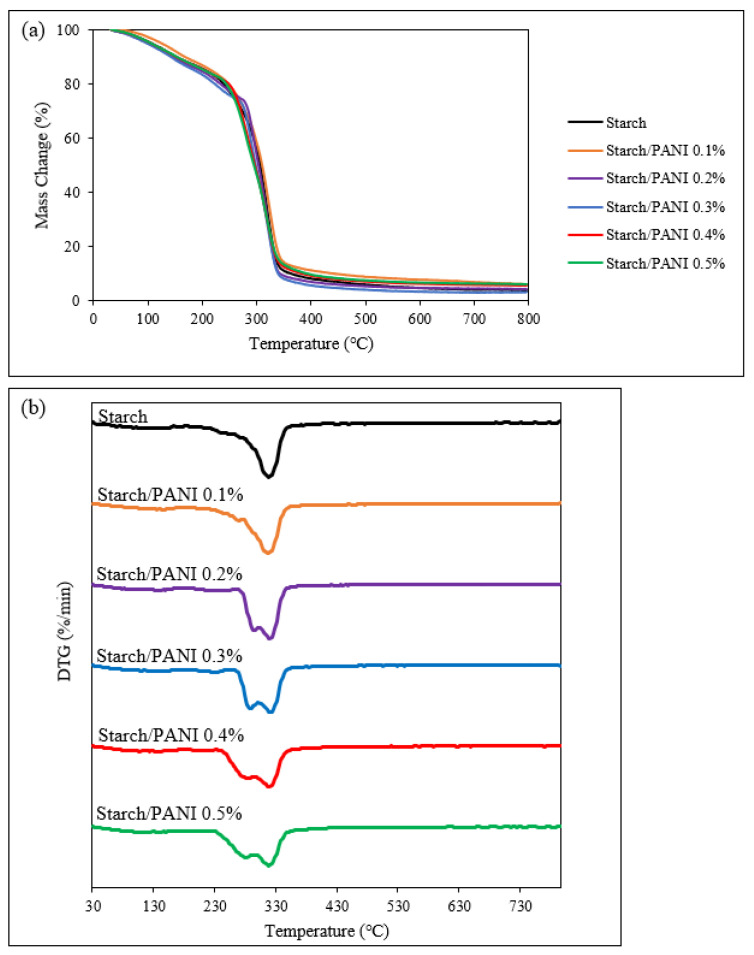
(**a**) TGA and (**b**) DTG thermograms of starch and starch/PANI (0.1%, 0.2%, 0.3%, 0.4%, 0.5%)-polymer films.

**Figure 7 polymers-14-01122-f007:**
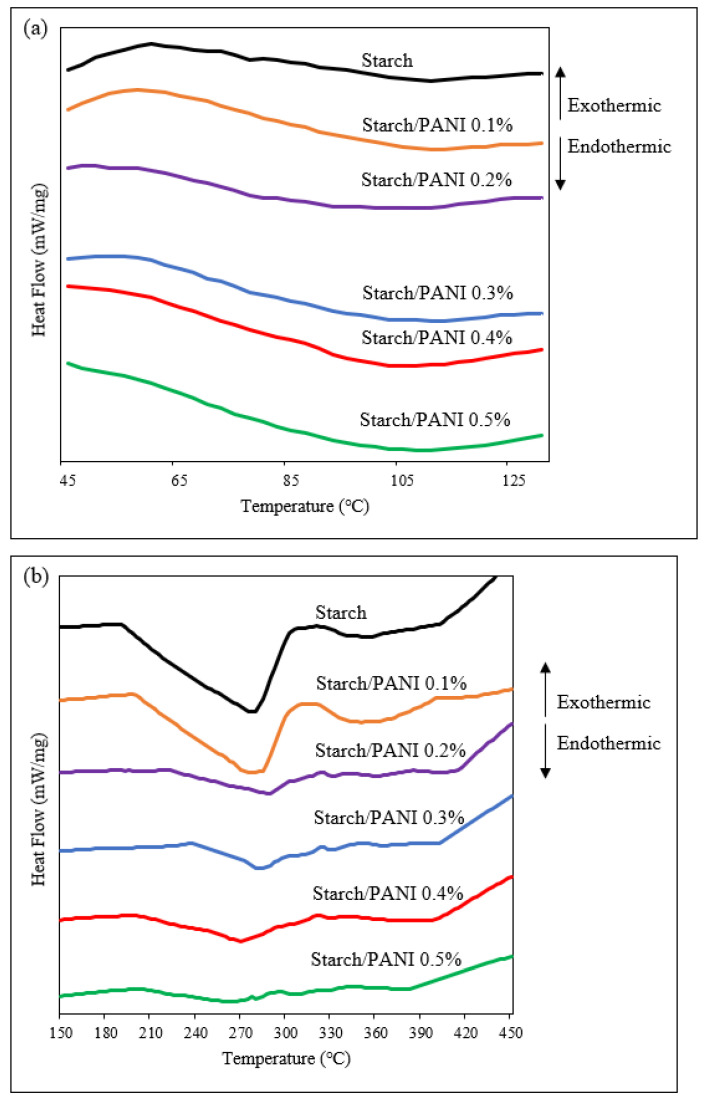
DSC thermograms of starch and starch/PANI (0.1%, 0.2%, 0.3%, 0.4%, 0.5%)-polymer films from (**a**) 45 °C to 130 °C and (**b**) 150 °C to 450 °C.

**Figure 8 polymers-14-01122-f008:**
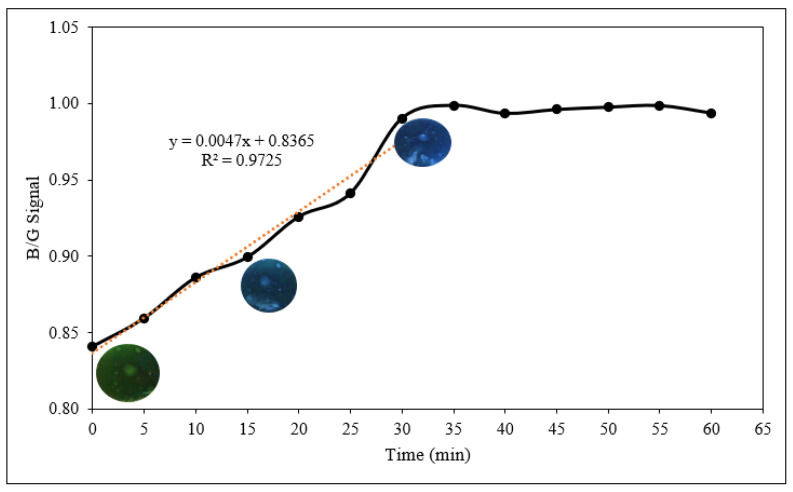
The response time of starch/PANI 0.4% film towards 600 ppm ammonia solution.

**Figure 9 polymers-14-01122-f009:**
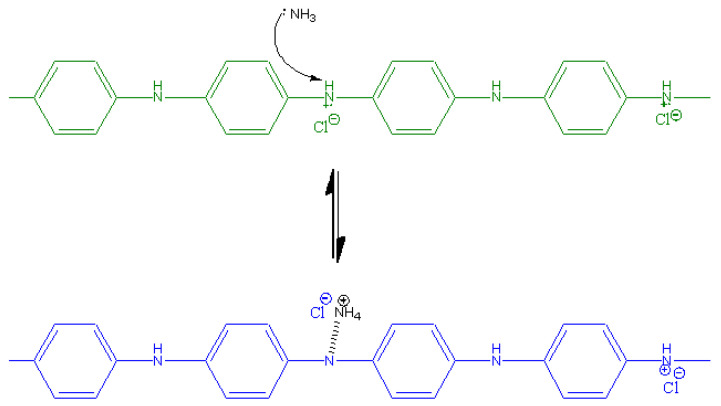
The interaction mechanism between PANI and ammonia.

**Figure 10 polymers-14-01122-f010:**
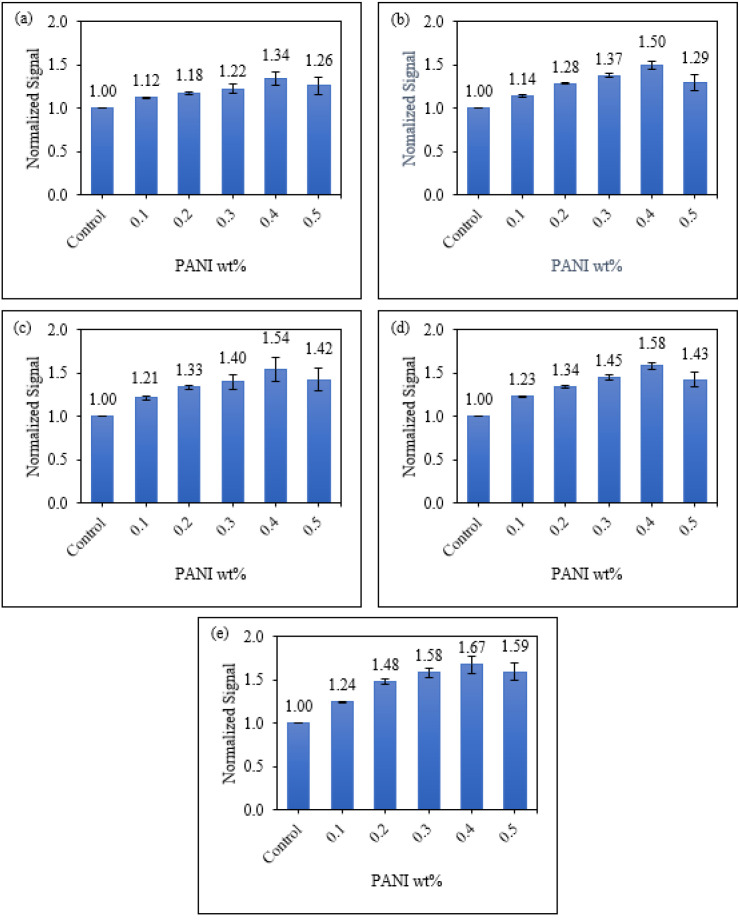
The normalised change of colour signal of control and starch/PANI (0.1%, 0.2%, 0.3%, 0.4%, 0.5%) films against (**a**) 200 ppm; (**b**) 400 ppm; (**c**) 600 ppm; (**d**) 800 ppm; (**e**) 1000 ppm ammonia solution.

**Figure 11 polymers-14-01122-f011:**
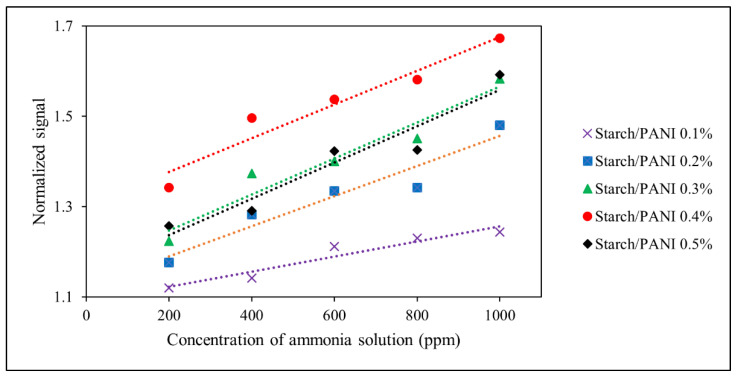
The colourimetric response of starch/PANI (0.1%, 0.2%, 0.3%, 0.4%, 0.5%) films against ammonia solution with varied concentration from 200 ppm to 1000 ppm.

**Figure 12 polymers-14-01122-f012:**
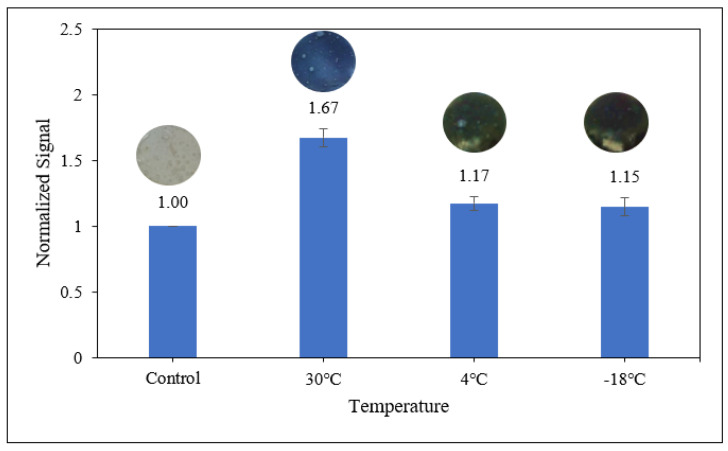
The colourimetric response of starch/PANI 0.4% to 1000 ppm ammonia solution at different temperatures (30 °C, 4 °C and −18 °C).

**Figure 13 polymers-14-01122-f013:**
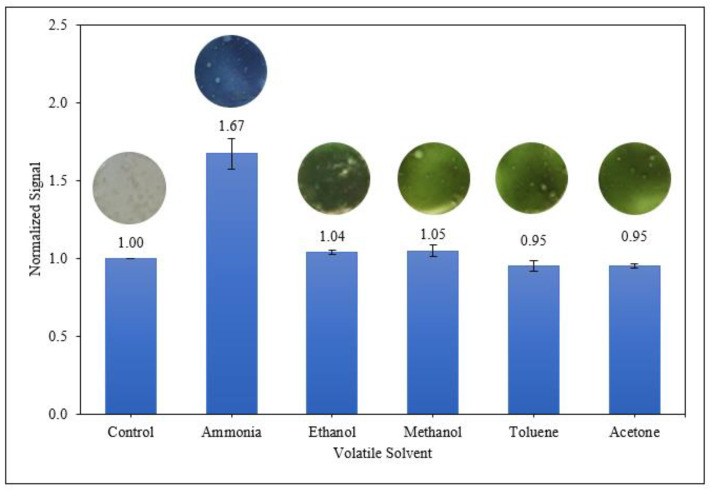
Colourimetric response of starch/PANI 0.4% film towards various volatile vapours.

**Table 1 polymers-14-01122-t001:** Degradation temperatures of starch and starch/PANI (0.1%, 0.2%, 0.3%, 0.4%, 0.5%)-polymer films.

Polymer Films	Temperature (°C)			
	Stage I	Stage II	Stage III	Stage IV
Starch	120.0	NIL	274.2	316.5
Starch/PANI 0.1%	140.1	NIL	292.4	318.7
Starch/PANI 0.2%	122.3	236.7	292.4	319.5
Starch/PANI 0.3%	128.0	228.8	287.5	319.3
Starch/PANI 0.4%	124.4	211.3	280.4	319.8
Starch/PANI 0.5%	116.5	189.6	277.5	317.8

**Table 2 polymers-14-01122-t002:** The glass-transition temperature (T_g_), melting temperature ™, and enthalpy of fusion (∆H_f_) of starch and starch/PANI (0.1%, 0.2%, 0.3%, 0.4%, 0.5%)-polymer films.

Polymer Films	T_g_ (°C)	T_m_ (°C)	∆H_f_ (J/g)
Starch	77.0	277.6	23.8
Starch/PANI 0.1%	77.0	281.3	19.5
Starch/PANI 0.2%	77.0	287.5	9.1
Starch/PANI 0.3%	76.3	282.5	6.3
Starch/PANI 0.4%	71.3	270.0	4.1
Starch/PANI 0.5%	68.4	262.6	3.2

**Table 3 polymers-14-01122-t003:** The colour changes of the starch/PANI films when exposed to the ammonia solution (800 ppm).

	Before	After
Starch/PANI 0.1%	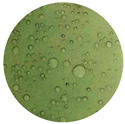	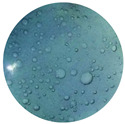
Starch/PANI 0.2%	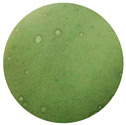	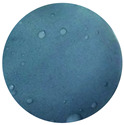
Starch/PANI 0.3%	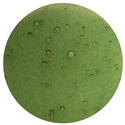	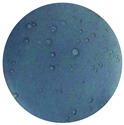
Starch/PANI 0.4%	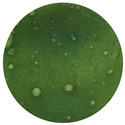	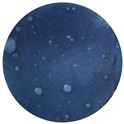
Starch/PANI 0.5%	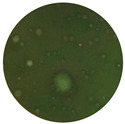	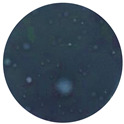

**Table 4 polymers-14-01122-t004:** Regression equation, correlation coefficient (R^2^), relative standard deviation (RSD), limit of detection (LOD), and limit of quantification (LOQ) of the analysis of starch/PANI (0.1%, 0.2%, 0.3%, 0.4%, 0.5%) films as ammonia sensor.

Polymer Films	Regression Equation		R^2^	RSD (%)	LOD (ppm)	LOQ (ppm)
	Slope	Intercept				
Starch/PANI 0.1%	0.0002	1.0891	0.9238 **	1.95	812	2459
Starch/PANI 0.2%	0.0003	1.1227	0.9208 **	2.38	166	502
Starch/PANI 0.3%	0.0004	1.1670	0.9200 **	6.10	148	448
Starch/PANI 0.4%	0.0004	1.3015	0.9359 **	8.72	145	440
Starch/PANI 0.5%	0.0004	1.1562	0.9379 **	12.41	167	505

** *p* < 0.01.

**Table 5 polymers-14-01122-t005:** The HSP (δ) values of the polymer matrix and volatile solvents [55,69].

Substrate	HSP (δ)/MPa^1/2^	∆δ with Starch (Dextran)/MPa^1/2^
Dextran	38.6	0.0
Glycerol	36.1	2.5
Ammonia	33.3	5.0
Ethanol	26.5	12.1
Methanol	29.6	9.0
Toluene	18.2	20.4
Acetone	20.0	18.6

## Data Availability

Most of the significant data collected and analysed were presented in this article. Additional data in this study are available on request from the corresponding author.

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
