# Peer review of "Starch/Polyaniline Biopolymer Film as Potential Intelligent Food Packaging with Colourimetric Ammonia Sensor"

_polymers, 2022, doi:10.3390/polym14061122_

Round 1
Reviewer 1 Report
This manuscript reported a method for preparation of PANI incorporated starch film, as well as the characterization of its thermal property and crystallinity. The performance of sensitivity as a colorimetric ammonia sensor was also evaluated systematically. The results demonstrated a potential application of PANI in smart food packaging.
Overall, the experiments are designed properly, the conclusions are supported by the characterizations. So, I would like to recommend it for Polymers.
Two comments:
Line 125, “After sonification, …”, I think that using “sonication” is better than “sonification”.
In Figure 9, the two structures of PANI are all in leucoemeraldine (white/colourless), and there is no emeraldine base (blue) structure. The authors should keep consistent with oxidation states (b, c) in Figure 1 in your manuscript.
Author Response
Dear Reviewer,
Thank you for your comments. Corrections has been done accordingly.
- Line 125: The term “sonification” was changed to “sonication”.
- The structure of emeraldine salt and base in Figure 1 is in bipolaronic form while in Figure 9 is polaronic form. Nevertheless, for better reader’s comprehension, Figure 1(b) and (c) was updated to the structure of emeraldine base and salt in polaronic form respectively.

Reviewer 2 Report
In this paper, the authors present starch/polyaniline (starch/PANI) biopolymer film was synthesized and investigated as an ammonia sensor for potential application as intelligent food packaging. FT-IR and XRD were used to confirm the composition of the biopolymer films, while UV-Vis spectrometry was applied to identify the oxidation state of PANI in emeraldine form. Which have some potential applications as colorimetric ammonia sensors for intelligent food packaging. This article is clear, concise, and suitable for the scope of the journal. Several small suggestions are supplied:
1. Suggest the authors improve Fig.1, some part is missing.
2. Suggest the authors give more detail about the normalized change of color singla of control and starch/PANI in sentences.
3. Suggest the authors enhance the introduction part about biopolymer with some lastest review such as:
Biocompatible and Biodegradable Polymer Optical Fiber for Biomedical Application: A Review Biosensors 11(12):472 2021.
Author Response
Dear Reviewer,
Thank you for your suggestions. Amendments were made as below:
- The structure of emeraldine salt and base in Figure 1 is in bipolaronic form while in Figure 9 is polaronic form. Therefore, for better reader’s comprehension, Figure 1(b) and (c) was updated to the structure of emeraldine base and salt in polaronic form respectively.
- Details regarding the normalized signals were included in the methodology section.
Line 157 – 160: “The sensitivity of the sensor is monitored based on the changes of colour signals mentioned in Equation 1 where the normalized change of colour signal was obtained by dividing the change of colour signal of the sample by the change of colour signal of control sample.”
- The latest review paper recommended by the Reviewer was cited in the introduction as Ref [6].
Line 34 – 36: “Starch and cellulose are biopolymers that have attracted the most attention, due to their abundance, biocompatibility, non-toxicity and simplicity of manipulation [5–7].”
[6] Wang, Y.; Huang, Y.; Bai, H.; Wang, G.; Hu, X.; Kumar, S.; Min, R. Biocompatible and Biodegradable Polymer Optical Fiber for Biomedical Application: A Review. Biosensors. 2021, 11 (12), 472. https://doi.org/10.3390/bios11120472.

This manuscript is a resubmission of an earlier submission. The following is a list of the peer review reports and author responses from that submission.
Round 1
Reviewer 1 Report
The article describes the development of ammonia sensors based on PANI and starch as polymeric matrix. The article includes the characterization of thermal and structural properties of the films as well the determination of the sensor capacity of the films. The article is well structured and deserves to be published in polymers after a minor revision. Particularly, Introduction should be improved incorporating a better revision about the use of PANI as sensor in intelligent food packaging.
Reviewer 2 Report
Dear Authors,
Please find my comments in the enclosed file.

Reviewer 3 Report
The manuscript presents a starch/polyaniline (PANI) biopolymer film used as a colorimetric ammonia sensor that can be used in intelligent food packaging. Despite a very extensive part describing changes in the structure and physicochemical properties of the film concerning the various quantitative addition of polyaniline to starch, the research on the evaluation of the analytical properties of the biofilm is carried out very superficially and with due diligence. The change in the color of the starch / PANI biofilm was largely dependent on the chemical composition of the film and the conditions under which the experiments were conducted.The degree of color change depended on the humidity and temperature of the environment.
The authors did not attempt to conduct experiments with a perishable food product that releases volatile amine compounds (not only ammonia) into the environment. Such tests would indicate the actual suitability of the tested biofilm for use in intelligent packaging, because in the real application of the indicator of freshness (spoilage) of food, the sensitivity of the indicator is very important and it may also depend on the way the sensor is placed in the packaging, the way of propagation of volatile substances in the packaging, the degree of adsorption analyte by indicator, etc. The term used in the conclusions that the presented index enables the detection of food spoilage is not confirmed by the research results.
The analytical properties of the ammonia sensor are very imprecise. The ammonia detection limit and the linearity of the signal with respect to the ammonia concentration are shown for very specific assay conditions. What was the repeatability of the obtained results? The photos show that the color of the indicator depended on its thickness (local highlights and shadows are visible). Did the Authors take into account the heterogeneity of the indicator discoloration in different places?
Finally, it is clear that how the indicator works depends on humidity and temperature. Have the authors investigated the durability of the presented indicators? How did they behave after longer storage? Were they not drying up and losing their activity? This is an important parameter in terms of the use of biofilm in intelligent packaging.